# AR-1-TO-3: SINGLE IMAGE TO CONSISTENT 3D OBJECT GENERATION VIA NEXT-VIEW PREDICTION

## ABSTRACT

Represented by Zero123 series of works, recent advancements in single-view 3D generation research have shown prominent progress by utilizing pre-trained 2D diffusion generation models. These approaches either generate multiple discrete views of a 3D object from a single-view image and a set of camera poses or produce multiple views simultaneously under specified camera conditions. However, it is hard to maintain consistency across different views and camera angles, especially for poses with large differences. In this paper, we introduce **AR-1-to-3**, a novel paradigm to generate multi-view images according to the input single image with significantly improved consistency in details. We achieve this by designing a novel auto-regressive scheme where novel views are generated based on previous views. The core of our method is first to generate views closer to the input view, which is utilized as contextual information to prompt the generation of farther views. To this end, we propose two image conditioning strategies, termed as **Stacked-LE** and **LSTM-GE**, to encode the sequence views. Particularly, Stacked-LE encodes the previously generated views into a stack embedding, which is employed as a local condition to modify the key and value matrices of the self-attention layers for denoising the target views of the current step. Meanwhile, LSTM-GE divides the previously generated views into two groups based on their elevations, whose feature vectors are encoded by two LSTM modules into high-level semantic information for global conditioning. Extensive experiments on the Objaverse dataset show that our method can synthesize more consistent 3D views and produce high-quality 3D assets that closely mirror the given image. Code and pre-trained weights will be made publicly available.

## 1 INTRODUCTION

Synthesizing 3D objects from a single image has long been a challenging problem in both computer vision and graphics communities. This task not only involves reconstructing the visible regions of a 3D object but also necessitates extrapolating the invisible parts. Inspired by the remarkable success of diffusion models in 2D image generation (Rombach et al., 2022; Peebles & Xie, 2023; Li et al., 2024; Zhou et al., 2024), numerous studies have sought to leverage the powerful 2D generative priors learned from large-scale image datasets and transfer this success to 3D generation (Qian et al., 2023; Long et al., 2024). For example, Zero123 (Liu et al., 2023a) pioneers the synthesis of novel views based on camera poses by fine-tuning pre-trained diffusion models on *(input_view, camera_pose, output_view)* triplets rendered from 3D objects on the Objaverse dataset (Deitke et al., 2023). Many subsequent works, such as Consistent123 (Lin et al., 2023), One-2-3-45 (Liu et al., 2024b), Cascade-Zero123 (Chen et al., 2023), also adhere to this paradigm.

Considering the independence of new views specified by different cameras, Zero123++ (Shi et al., 2023) proposes to merge all target views into a grid image to jointly model the distribution of multiple views. Nevertheless, these methods may produce multi-views inconsistent with the input image, resulting in unsatisfactory synthesis quality of 3D objects, particularly when the texture information of the 3D objects is complex and the gap between camera poses is large. Fig. 1 illustrates some examples of these two paradigms of methods in this scenario.

How do humans imagine the 3D shape and appearance of an object given a single camera view? In addition to leveraging a wealth of accumulated prior knowledge, we typically also refer to the

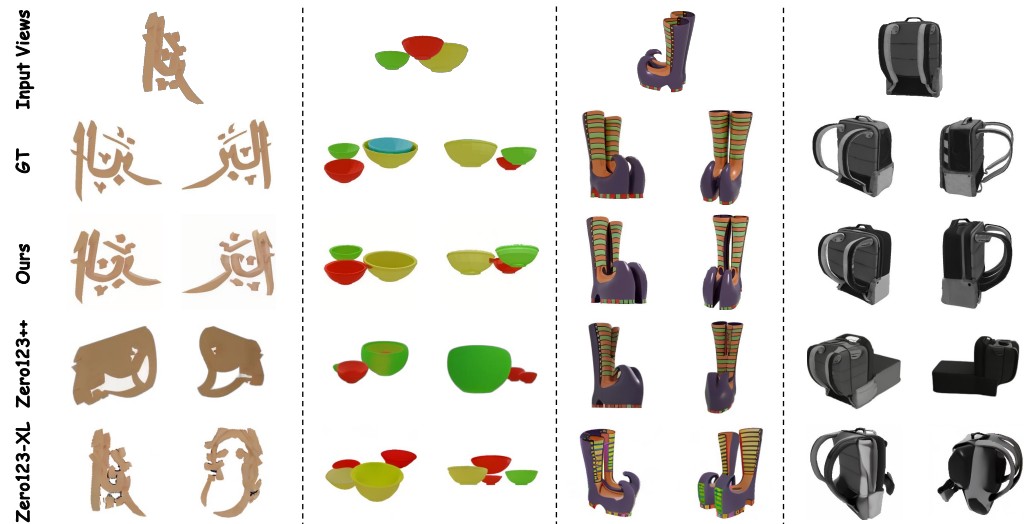

Figure 1: Examples of the three different paradigms for multi-view generation.

contextual information of the current object (Cohen et al., 1990). To be specific, we initially infer the views of objects closer to the given viewpoint, which are employed as context to restore the full view of the 3D object from near to far. Motivated by this, we rethink the way of generating consistent views and present **AR-1-to-3**, a novel paradigm that progressively generates all target views in an autoregressive fashion.

To be specific, we follow the $3 \times 2$ grid image generation strategy proposed by Zero123++ (Shi et al., 2023), in which the 6 target camera poses consist of interleaving elevations of $20°$ downward and $-10°$ upward, combined with azimuths that start at $30°$ and increase by $60°$ for each pose. Given a source image, AR-1-to-3 starts by generating the first row views of the $3 \times 2$ grid, and gradually generates the remaining row views in the subsequent steps with a fixed row azimuth interval of $120°$ for each step. At each step, the views of two different elevations can exchange information, and the views generated in the previous steps are utilized as conditions to generate the views for the current step. To this end, we develop a conditional feature encoding scheme for both the image local conditioning and global conditioning of the diffusion models. For the former, we design a **Stacked-LE** manner where the features of previously generated views encoded by the denoising UNet model are stacked to the key and value matrices of the self-attention layers when denoising the current views. For the latter, we propose a **LSTM-GE** strategy, in which the features of views already generated encoded by the CLIP (Radford et al., 2021) model are fed to two branches of LSTM according to their elevations. The output features are concatenated together for the cross-attention layers during the denoising process of the current views.

We evaluate the performance of AR-1-to-3 on a wide range of 3D objects collected from the Obja-verse dataset (Deitke et al., 2023). By introducing the autoregressive manner coupled with the pro-posed Stacked-LE and LSTM-GE strategies into multi-view latent diffusion models, our AR-1-to-3 enables the generation of consistent and accurate 3D views for various objects. The experimental results demonstrate that our AR-1-to-3 is able to produce high-quality 3D assets and outperform other state-of-the-art image-to-3D methods. We hope that the auto-regressive scheme provided by this paper will inspire future works and our AR-1-to-3 could become a base multi-view generative model for the 3D generative AI community.

To sum up, our main contributions can be summarized as follows:

- We present a novel autoregressive framework, termed as AR-1-to-3. It takes a single-view image as input to progressively generate consistent target views from near to far.

- We develop the Stacked-LE and the LSTM-GE strategies to encode the already generated views for the local conditioning and global conditioning of the multi-view diffusion models.

- Extensive experiments on the large-scale 3D dataset, *i.e.,* Objaverse, demonstrate that our approach could generate more consistent 2D multi-view images than previous works and produce high-quality 3D assets.

## 2 RELATED WORK

### 2.1 2D DIFFUSION MODELS FOR 3D GENERATION

Diffusion Models (Ho et al., 2020; Rombach et al., 2022) pre-trained on large-scale 2D datasets have demonstrated remarkable performance in generating high-quality images and powerful zero-shot zero-shot generalization. In recent years, a significant amount of effort has been consecutively devoted to transferring the strong priors of 2D diffusion models to 3D generation. For example, DreamFusion (Poole et al., 2022) and SJC (Wang et al., 2023) propose to distill the knowledge of a pre-trained 2D diffusion model by feeding the rendered views to it and performing per-shape optimization. However, these methods are typically extremely time-consuming and suffer from artifacts such as over-saturated colors and the "multi-face" problem. Zero123 (Liu et al., 2023a) pioneers an open-world single-image-to-3D framework, in which diffusion models are fine-tuned to synthesize new views conditioned on an input view and a set of discrete camera poses. Many subsequent works employ Zero123 as a base module to generate 3D objects. Magic123 (Qian et al., 2023) combines the 3D prior of Zero123 and the 2D prior of the stable diffusion model together to enhance the quality of generated 3D meshes. One-2-3-45 (Liu et al., 2024b) uses Zero123 to generate multi-view images, which are lifted to 3D space to assist in the generation of 3D meshes. There also have been several approaches, *e.g.,* Consistent123 (Lin et al., 2023) and Cascade-Zero123 (Chen et al., 2023), proposed to improve Zero123 by incorporating extra priors, like boundary and redundant views. Besides, more and more attention has been drawn to enforce consistency between the generated multiple views. SyncDreamer (Liu et al., 2023b) adopts a 3D-aware attention mechanism to correlate the corresponding features across different views. MVDiffusion (Tang et al., 2023) generates multi-view images in parallel through the weight-sharing multi-branch UNet with shared weights and correspondence-aware attention. More recently, Zero123++ (Shi et al., 2023) proposes a strategy of tiling six target views surrounding the 3D object into a grid image to simultaneously generate multiple views and enable the correct modeling of their joint distribution. This strategy also has been successively adopted by follow-up works like One-2-3-45++ (Liu et al., 2024a), Instant3D (Li et al., 2023), and InstantMesh (Xu et al., 2024), *etc*.

In contrast to these methods heavily relying on the 2D priors of diffusion models, our AR-1-to-3 inspired by human thinking also pays attention to the contextual information of the current object. The method most similar to our method is Cascade-Zero123 (Chen et al., 2023). It first uses a multi-view diffusion model to generate many extra views, which, along with the input image, are then fed into another diffusion model to produce a specific target view. Different from this method, our AR-1-to-3 takes into account the relationship between the target views and the input image, and utilizes a diffusion model to establish the potential sequence between them. Given the target views closer to the input image are easier to generate, AR-1-to-3 gradually generates all target views from near to far in an autoregressive manner, in which the views generation of each step could be assisted by the views generated in earlier steps.

### 2.2 AUTOREGRESSIVE VISUAL GENERATION

The autoregressive scheme, which is used for analyzing and predicting time series data, relies on the idea that the current value in a series can be explained by its previous values. Based on this scheme, many researchers propose a series of classic sequential modeling methods, *e.g.,* LSTM (Hochreiter, 1997), ConvLSTM (Shi et al., 2015), Transformer (Vaswani, 2017), which have excelled in various communities including natural language processing, computer vision, and multi-modal learning. In recent years, there has been a growing interest in applying such an autoregressive manner to visual generation. VQVAE (Van Den Oord et al., 2017) revolutionizes the learning of discrete representations by incorporating the codebook mechanism, enabling efficient encoding and decoding processes of images. VQGAN (Esser et al., 2021) adopts a transformer architecture to model serialized visual parts and introduces adversarial loss during the training process. Parti (Yu et al., 2022) proposes a pathways autoregressive model treating the image generation as a sequence-to-sequence modeling to generate high-fidelity photorealistic images. LlamaGen Sun et al. (2024) and VAR (Tian et al., 2024) scale image generation by incorporating multimodal large models.

In this work, we observe there is also a potential sequential nature among the target views, which can be divided into several steps with equidistant camera pose intervals. As a result, we propose to generate multiple novel views in an autoregressive manner.

## 3 METHODOLOGY

### 3.1 PRELIMINARIES

We introduce the preliminaries of Zero123++ (Shi et al., 2023), the base multi-view latent diffusion model adopted in our work, which is beneficial for understanding the designs in AR-1-to-3.

**Multi-View Generation.** To generate consistent multi-view images, Zero123++ proposes tiling six target views with a $3 \times 2$ layout into a single grid frame to model the joint distribution among them. For the target views, a fixed set of relative azimuth and absolute elevation angles are adopted in this work. Specifically, the six camera poses contain alternating elevation angles of $20°$ downwards and $-10°$ upwards, along with azimuth angles starting from $30°$ relative to the input azimuth angle and incrementing by $60°$ for each subsequent pose.

**Stable Diffusion.** Zero123++ chose Stable Diffusion (SD) as the generative model since it is open-sourced and has been trained on various internet-scale image datasets. The geometric priors that SD learns about natural images are utilized for novel view synthesis under the image and camera conditions. SD performs the diffusion process within the latent space of a pre-trained autoencoder whose encoder and decoder are denoted as $\mathcal{E}(\cdot)$ and $\mathcal{D}(\cdot)$, respectively. At the diffusion time step $t$, the objective for fine-tuning the denoiser UNet $\epsilon_\theta(\cdot)$ can be formulated as:

$$\mathcal{L} = \mathbb{E}_{z \sim \mathcal{E}(x), y, t, \epsilon \sim \mathcal{N}(0, I)} ||\epsilon - \epsilon_\theta(z_t, t, c_\theta(y))||_2^2, \tag{1}$$

where $x$ is the target grid image which is encoded and perturbed to a feature with Gaussian noise $\epsilon$, *i.e.,* $z_t$, and $c_\theta(y)$ represents the embedding encoded from the condition image $y$.

**Image Condition.** The image conditioning techniques employed in Zero123++, *i.e.,* $c_\theta(y)$, can be divided into two aspects, *i.e.,* local condition and global condition. The local conditioning strategy is tailored for the pixel-wise spatial correspondence between the input and the target views. To this end, Zero123++ adopts a variant of the Reference Attention operation (Zhang, 2023), which runs the denoising UNet on the input image and appends the self-attention key and value matrices from it to the corresponding attention layers when denoising the target views. As far as the global conditioning mechanism, the CLIP text embedding of an empty text is added with the CLIP image embedding of the input image multiplied by a trainable set of global weights to provide high-level semantic information for the cross-attention of the denoising UNet.

### 3.2 AR-1-TO-3

Existing methods either generate multiple discrete viewpoints from a single input view and a set of camera poses, like Zero123 (Liu et al., 2023a), One-2-3-45(Liu et al., 2024b), or simultaneously generate multiple views in a grid layout based on specified camera conditions, such as Zero123++ (Shi et al., 2023), Instant3D (Li et al., 2023). Despite achieving excellent performance in many scenarios, these methods are still prone to generating several target views that exhibit geometric and textural inconsistencies with the input image. We argue that the underutilization of contextual information of current objects during the generation process should be responsible for this issue.

The core of this work is to progressively generate target views in an autoregressive fashion so that the closer views generated earlier can be used as supplementary information for the generation of the farther views. Fig. 2 shows the end-to-end architecture of our AR-1-to-3. In this work, we follow the paradigm of Zero123++ generating 6 specific camera conditional target views. In contrast, our method generates these views step by step rather than all at once. Besides, Zero123++ has demonstrated that generating multiple target views simultaneously contributes to accurately modeling the joint distribution of these views. Therefore, each step in our generative strategy refers to a row of the $3 \times 2$ layout, containing two target views with different elevations and $60°$ difference in azimuth. Moreover, the camera pose difference between steps is a fixed azimuth angle of $120°$, which is suitable for the scheme of next-views generation. As a result, the different target views at each step can exchange information, and the target views generated in the previous steps can be utilized as extra conditions to generate the views for the current step.

We achieve such autoregressive generation by designing two image conditional strategies that encode sequence view information to fine-tune the denoising UNet model. These two strategies, denoted as Stacked Local Feature Encoding (Stacked-LE) and Long Short-Term Global Feature Encoding (LSTM-GE), correspond to the local and global image conditioning techniques in Zero123++,

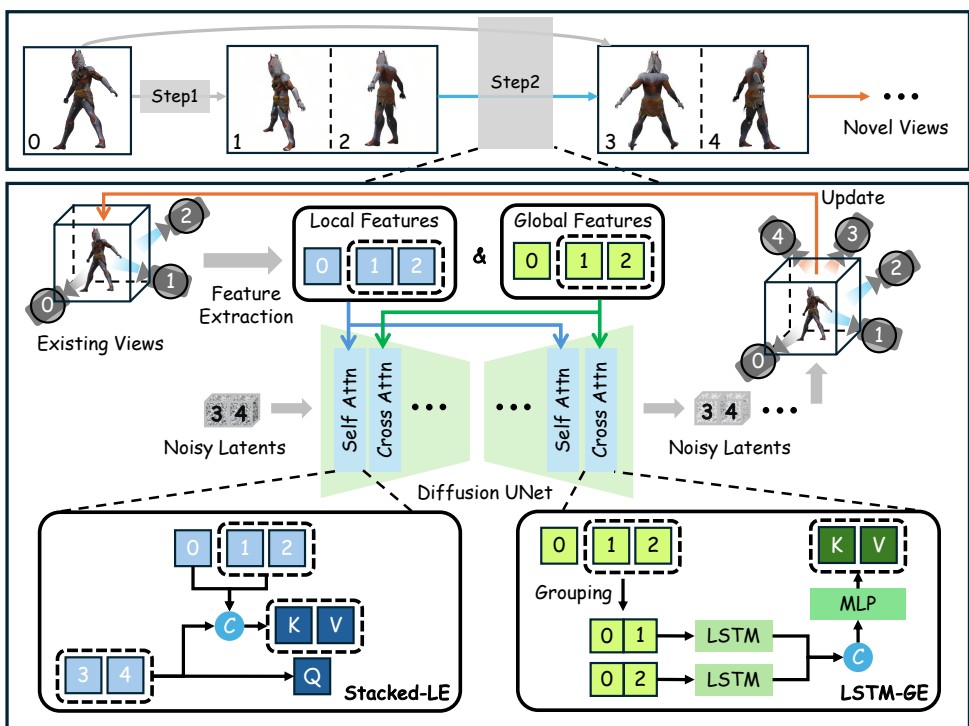

Figure 2: Pipeline of the autoregressive process in our AR-1-to-3. Starting from the input image, our method generates all target views incrementally from near to far, with each step conditioned on the existing views from previous steps. To achieve this, Stacked-LE and LSTM-GE strategies are proposed to encode the local features and global features of the partial view sequence as the image conditions of the denoising UNet.

respectively. The optimization objective can still be represented by Eqn. 1, and we will elaborate on our image conditional policy, *i.e.,* $c_\theta(y)$, in the subsequent Sec. 3.3 and Sec. 3.4.

Through multi-step autoregression, our AR-1-to-3 gradually generates 6 target views, which are fed to a sparse-view large reconstruction model to obtain a 3D object. In this work, we choose the pre-trained InstantMesh (Xu et al., 2024) as our 3D reconstruction model, which encodes the multi-view images as triplane features via a transformer architecture and predicts the point color and density for volumetric rendering by a multi-layer perceptron.

### 3.3 STACKED LOCAL FEATURE ENCODING

In this subsection, we introduce how our method encodes the latent features of the input image and the generated partial target-view sequence as local conditions for the operation of Reference Attention to generate the target views of the current step. Note that the denoising UNet model is a multi-level architecture and the hidden dimensions may vary across different self-attention layers. It is challenging to encode the latent features of the condition view sequence at these positions using a single network. Considering that these reference features and the attention representations in self-attention layers come from the same positions in the U-Net network, sharing the same spatial and channel dimensions, we naturally thought of encoding them into a unified representation by stacking them along the spatial dimension. This strategy offers two significant advantages: 1) it can encode any number of reference features at the self-attention layer. 2) it allows the reference features to be directly fed into the attention module, enabling the reuse of weight parameters without any additional design required. We term this local feature encoding strategy as Stacked-LE whose details are shown in the bottom left corner of Fig. 2.

Formally, at the $k$-th step of autoregression, with a total of $2k-1$ reference views, we aim to predict the $(2k)$-th and $(2k+1)$-th target views, where the variable $k$ ranges from 1 to 3. We first feed the reference views into the denoising UNet model separately and record their key/value matrices at

the self-attention module. Gaussian noise at the same level as the denoising input is added to the reference image so that the UNet can focus on the relevant features for denoising at the current noise level. Then, we perform another forward pass with the U-Net to denoise the target views of the current step. During this process, the records in each layer are stacked together to modify the key and value matrices in the self-attention module of the corresponding layer, which can be defined as:

$$s_i^* = \text{Concat}([e_i^1, e_i^2, ..., e_i^{2k-1}, s_i]), \tag{2}$$

where $s_i \in \mathbb{R}^{B \times L \times D_i}$ denotes the key or value matrices of the $i$-th self-attention, and $e_i^j$ is the recorded embeddings of the $j$-th reference view. Note that $B$, $L$ and $D_i$ are the batch size, token number, and dimension of the key /value matrices. Further, we compute the self-attention as follows:

$$O_i = \text{Attention}([Q_i, K_i^*, V_i^*]), \tag{3}$$

It is noteworthy that Stacked-LE may encounter GPU burden issues when the sequence length becomes excessively long. To alleviate this concern, we introduce a novel random sampling strategy inspired by recent works on video generation (Zhou et al., 2024). Specifically, we retain all $L$ tokens of $e_i^1$ as they originate from the input image, and perform random sampling with a proportion of $\alpha$ on the $L$ tokens from each generated target view. When the $\alpha$ value is 0, Stacked-LE degenerates into the original Reference attention; when $\alpha$ is 1, it is equivalent to using all tokens as in Eqn. 2. As far as the selection of $\alpha$, please refer to Sec. 4.4 for more details.

### 3.4 Long Short-Term Global Feature Encoding

In this subsection, we present the details of encoding the CLIP features of the input images and the existing target-view sequence as global conditions, which provide high-level semantic information via cross-attention to generate the target views of the current step. We empirically observe that the CLIP features of the conditional view sequence are vectors with the same channel dimension and 1D spatial dimensions. Besides, they can be divided into two sub-sequences with an azimuthal angle spacing of $120°$ based on their elevation angles. Thus, it is well-suited to process such sequences with the Long Short-Term Memory (LSTM) Network (Hochreiter, 1997). Furthermore, this manner has two interesting merits, mitigating computational burden: 1) It can encode feature vectors from the conditional views into two vectors, regardless of the number of views. 2) The LSTM structure possesses a strong ability to model sequential representations while requiring fewer parameters. This strategy is named as LSTM-GE, and the details are shown in the bottom left corner of Fig. 2.

Given $2k$-1 conditional views at the $k$-th step of autoregression, we first send them to the image encoder of the CLIP model for their visual features, which are represented as $F \in \mathbb{R}^{B \times (2k-1) \times D}$. Then, we partition these features into two groups according to the elevation angles of their respective views. Note that the feature of the input image is a special case, which is included in both groups. We denote the two grouped features as $F_1 \in \mathbb{R}^{B \times k \times D}$ and $F_2 \in \mathbb{R}^{B \times k \times D}$, and feed them to two separate LSTM modules. Next, the hidden states of the $k$-th step of the two LSTM modules, *i.e.*, $h_l^k$, are selected as their respective outputs, *i.e.*, , $I_l$. This process can be formulated as follows:

$$I_l = \text{LSTM}_l([F_l, (h_l^0, c_l^0)]), \quad l \in \{0, 1\} \tag{4}$$

where $h_l^0$ and $c_l^0$ are the hidden state and cell state, respectively, both initialized as zero vectors. Finally, the outputs of the two LSTM modules are concatenated together along the channel dimension, followed by a MLP layer and a trainable set of global weights $W \in \mathbb{R}^{77 \times 1}$, to form global embeddings for the cross-attention of the denoising UNet:

$$T = W \cdot \text{MLP}(\text{Concat}([I_0, I_1])), \quad T \in \mathbb{R}^{B \times 77 \times D}. \tag{5}$$

Note that we remove the CLIP embedding of the empty text in the original global conditioning, as it has little impact on our final results.

## 4 Experiments

### 4.1 Experiment Setup

**Datasets.** We conduct experiments on the Objaverse (Deitke et al., 2023) dataset, which is the most popular large-scale open-source 3D Dataset with about 800k annotated mesh objects. We pick

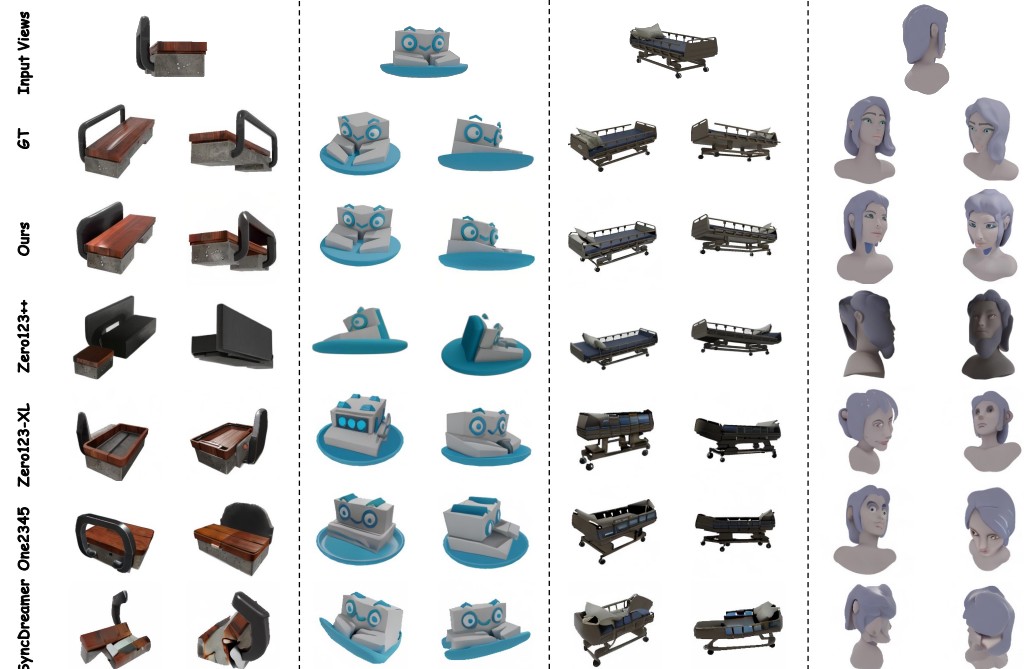

Figure 3: Visual comparisons of the novel views synthesized by our AR-1-to-3 and recent popular methods on multi-view generation. Compared with the existing approaches, the new views from our AR-1-to-3 are more consistent with both each other and the input views.

out 300 objects covering various categories, *e.g.,* cartoon characters, furniture, multi-objects, and word-objects, *etc*, to evaluate the performance of the methods. The remaining samples are used for training. Following the protocol of Zero123++ (Shi et al., 2023), we render 7 images for each object, including an input image and 6 target images. To be specific, an elevation angle ranging from $-20°$ to $45°$ and an azimuth angle ranging from $0°$ to $360°$ are randomly sampled to render the input image. The camera poses of the 6 target images involve interleaving absolute elevations of $20°$ and $-10°$, paired with azimuths relative to the input image that start at $30°$ and increase by $60°$ for each pose. Besides, all rendered images are set to a white background to ensure that the diffusion model produces images of this nature, thereby avoiding the trouble of removing the background when reconstructing 3D objects. We will open-source these rendered images in our project.

**Evaluation.** We evaluate the performance of the methods by comparing the novel views generated by the multi-view diffusion models or rendered from the synthesized 3D meshes with the ground truth views. Following the common settings for image comparison (Lin et al., 2023; Chen et al., 2023; Xu et al., 2024), we report four popular metrics, including Peak Signal-to-Noise Ratio (PSNR), Perceptual Loss (LPIPS) (Zhang et al., 2018), Structural Similarity (SSIM) (Wang et al., 2004), and CLIP-score (Radford et al., 2021).

**Implementation Details.** We train AR-1-to-3 on the render images from about 800k objects of the Objaverse dataset for 150k steps with a total batch size of 32 on 8 NVIDIA A100 (80G) GPUs. The learning rate is initialized as $1e$-5 and changes every 25k steps in a cycle, along with the AdamW optimizer (Loshchilov & Hutter, 2019) and CosineAnnealingWarmRestarts scheduler (Loshchilov & Hutter, 2016). We randomly select a $k$ from $\{1, 2, 3\}$ to build the autoregressive pattern, where the first $2k$-1 views form the conditional images and the following two views make up the target views. We resize the size of the conditional images to a value between $128$ and $512$ so that the model is capable of adjusting to different input resolutions and producing more clear images. Meanwhile, we resize each target view to 320, thereby the size of the grid image during the autoregressive process is $320 \times 640$. Besides, we employ the linear noise schedule and v-prediction loss in Zero123++ (Shi et al., 2023) rather than the alternatives in the Stable Diffusion model (Rombach et al., 2022). During the inference stage, starting from the input image, our AR-1-to-3 generates all target views in three steps, with each step generating the current target views based on the existing views.

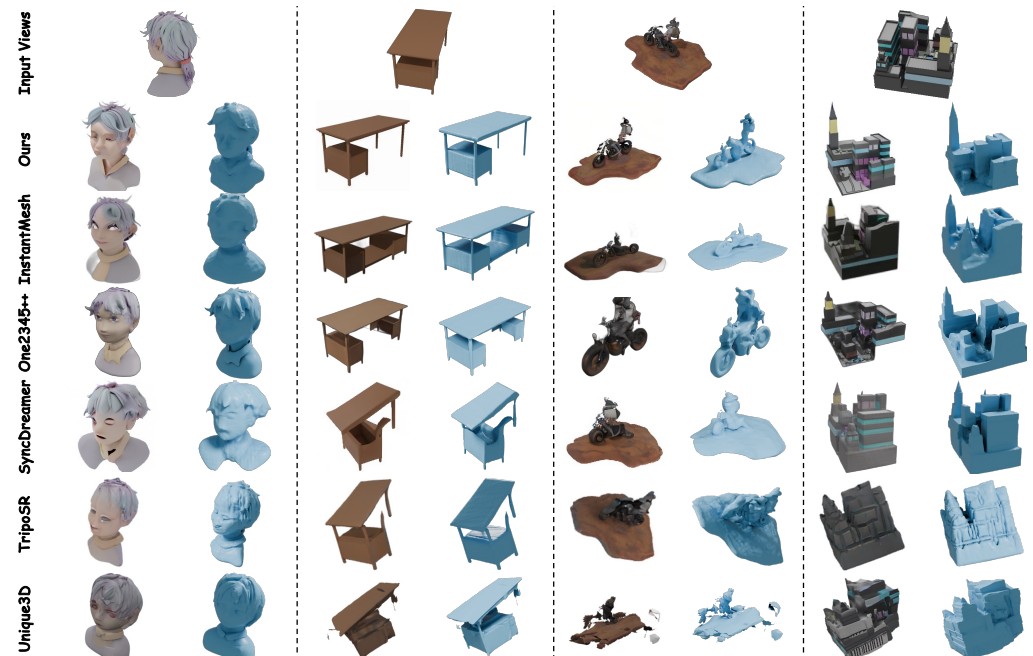

Figure 4: Visual comparisons between our AR-1-to-3 and recent cutting-edge methods on single-view image to 3D object generation. Note that the 3D results of Unique3D, TripoSR, and One2345++ are obtained by sending the input views to their official demos on Huggingface.

## 4.2 QUALITATIVE RESULTS

We first conduct visualization experiments on novel view synthesis and image-to-3d using a wide range of 3D objects to verify the effectiveness of our AR-1-to-3. To highlight the advantages of our method in contextual reasoning and zero-shot generalization, we adopt the input views with a certain degree of offset relative to the frontal views of the 3D samples.

**Novel View Synthesis.** Fig. 3 shows the visual comparisons of the synthesized views between our AR-1-to-3 and recent popular methods in multi-view generation, including Zero123++ (Shi et al., 2023), Zero123-XL (Liu et al., 2023a), One-2-3-45 (Liu et al., 2024b), SyncDreamer (Liu et al., 2023b). Note that Zero123-XL denotes an enhanced Zero123 model pre-trained on the Objaverse-XL dataset, and the open-source One-2-3-45 project also employs this version of Zero123 to generate the 8 views for its first stage. We utilize the elevation estimation implements of One-2-3-45 for the necessary elevation estimation procedures in Zero123-XL and SyncDreamer. Among these difficult-to-maintain consistency scenes, some existing methods produce multiple inconsistent novel views, as shown in the bench results of Zero123++ and the cartoon figure of Zero123-XL. Some approaches may even be confused and generate new views that differ significantly from the input image, as shown in the bench predictions of SyncDreamer and the four-wheeled beds of One-2-3-45. In contrast, our AR-1-to-3 is able to capture texture details of 3D objects and synthesize consistent and high-quality multi-view images, which is attributed to the full utilization of contextual information.

**Single Image to 3D.** We compare our AR-1-to-3 with recent open-source image-to-3d methods, *i.e.,* InstantMesh (Xu et al., 2024), One2345++ (Liu et al., 2024a), SyncDreamer (Liu et al., 2023b), and TripoSR (Tochilkin et al., 2024) and Unique3D (Wu et al., 2024), based on four images of different categories: a boy, a desk with a storage cabinet, a man riding a motorcycle on land, and a villa. For each mesh generated by these approaches, we visualize both the textured renderings (left) and pure geometry (right) on a novel view different from the input view. As depicted in Fig. 4, our AR-1-to-3 is capable of generating 3D meshes with consistent appearance and plausible geometry under limited input view information. Nevertheless, it is a real struggle for our counterparts to achieve this. For example, InstantMesh and One2345++ tend to generate an additional storage cabinet for the desk. We speculate that this is due to their excessive reliance on the symmetry prior of diffusion models during the generative process of new views, with less consideration for the contextual

information of the object itself. Although SyncDreamer does not generate additional components, the desk it produces exhibits significant geometric deformations. We find that the reason for this is the inconsistency among the 16 views generated by its diffusion model, which is employed for reconstructing the 3D object. Different from these approaches, our AR-1-to-3 effectively utilizes the contextual information of the object itself, progressively from nearby to distant, during the autoregressive generation of all target views. As a result, our method can achieve excellent performance in image to 3D object generation.

## 4.3 QUANTITATIVE RESULTS

We also quantitatively evaluate the performance of our AR-1-to-3 and other state-of-the-art methods using the selected 300 samples, which are excluded during the training process. For each 3D sample, these methods generate six target views according to the settings of Zero123++ (Shi et al., 2023). For the methods that do not support changing the elevation an-

Table 1: Quantitative comparison of novel view synthesis on the test split of Objaverse dataset.

| Model | PSNR ↑ | LPIPS ↓ | SSIM ↑ | CLIP-Score ↑ |
|---|---|---|---|---|
| Zero123-XL | 14.75 | 0.212 | 0.803 | 0.711 |
| Zero123++ | 14.83 | 0.201 | 0.815 | 0.718 |
| SyncDreamer | 14.88 | 0.198 | 0.817 | 0.723 |
| One-2-3-45 | 15.04 | 0.193 | 0.821 | 0.735 |
| AR-1-to-3 (Ours) | 20.28 | 0.121 | 0.857 | 0.887 |

gle, *e.g.,* SyncDreamer, we first generate 3D meshes following their default pipelines and then use Blender to render the views of these camera poses. As shown in Tab. 1, our AR-1-to-3 surpasses other cutting-edge methods across all metrics by a large margin. These results further demonstrate the superiority of our AR-1-to-3 over existing methods on the multi-view generation of 3D objects.

## 4.4 ABLATIVE STUDY

In this subsection, we conduct provide ablative studies and analysis to investigate the effectiveness of our designs in AR-1-to-3, including the sequence order in autoregressive generation, the sampling ratio in Stacked-LE, and the encoding strategy in LSTM-GE.

**Effect of Different Sequence Orders.** We define the sequence order from near to far in terms of camera poses relative to the input viewpoint as the normal order. We also provide two variants of the sequence order, *i.e.,* reverse order and random order, to generate all target views. To be specific, the reverse order refers to the tar-

Table 2: Ablative studies on the sequence order during the autoregressive generation.

| Model | PSNR ↑ | LPIPS ↓ | SSIM ↑ | CLIP-Score ↑ |
|---|---|---|---|---|
| Reverse | 20.19 | 0.124 | 0.851 | 0.882 |
| Random | 17.36 | 0.167 | 0.839 | 0.774 |
| Normal (Ours) | 20.28 | 0.121 | 0.857 | 0.887 |

get view sequence where the cameras move from far to near relative to the input view following the settings of Zero123++. The random order denotes the target view sequence that places the middle row of the $3 \times 2$ grid layout as the first position, followed by the remaining two rows. As shown in Tab. 2, our normal order achieves the best performance, while the random sequence performs the worst, demonstrating the effectiveness of modeling the target views in a sequential manner. It is noteworthy that the reverse order achieves a similar performance to our normal order. We believe the reason for this is that the reversed sequence can be seen as the camera moving from near to far in another direction, as the camera moves in a circular motion around the 3D object.

**Sampling Ratio of Latent Feature Sequence.** We investigate the impact of sampling ratio, *i.e.,* $\alpha$, on the performance and efficiency of our model in a train-free fashion. We freeze the trained model parameters and calculate the two measure values, *i.e.,* PSNR and FLOPS, at different sampling rates. As shown in Fig. 5, the PSNR value drops sharply when $\alpha$ is 0. We believe the reason is that the parameters of AR-1-to-3 have been optimized based on additional view features. If the model misses them, ambiguity may arise in understanding 3D objects. When $\alpha$ is greater than 0, the PSNR value enters a normal level and gradually increases with $\alpha$. Meanwhile, the FLOPS value at the max-

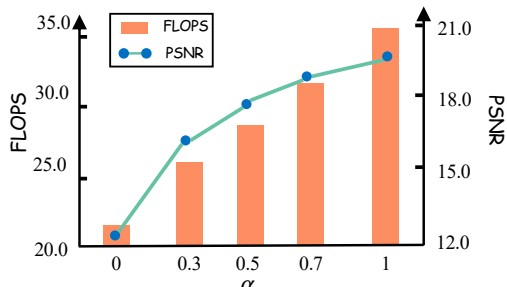

Figure 5: Ablative studies on Performance VS. Efficiency at different sampling ratios of the latent features.

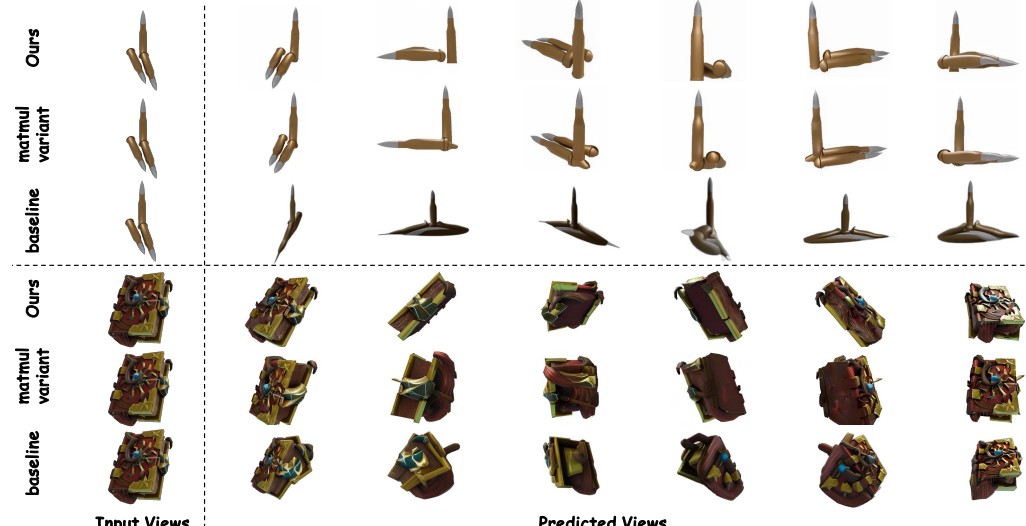

Figure 6: Ablative studies on the global feature encoding strategy of the conditional views. We present the 6 views of the $3 \times 2$ layout from the baseline model, the 'matmul' variant, and our AR-1-to-3. The superior performance of our approach in these samples demonstrates the effectiveness of LSTM-GE in high-level global semantic understanding for 3D objects.

sequence length continues to increase. In this paper, we set the $\alpha$ value to 1 by default. In practice, readers can choose a value between 0.3 and 1 according to their sequence length and GPU memory.

**Encoding Strategy of Global Feature Sequence.** Our LSTM-GE employs two LSTM modules to encode the global features of conditional view sequence. To highlight the effectiveness of our proposal, we propose a matrix multiplication ('matmul') variant to encode these features. Specifically, we stack these features into a matrix with shape $\mathbb{R}^{(2k-1) \times D}$. Meanwhile, we repeat the trainable weights in global condition $(2k\text{-}1)$ times to obtain a matrix with shape $\mathbb{R}^{77 \times (2k-1)}$. We multiply these two matrices to obtain a new matrix with shape $\mathbb{R}^{77 \times D}$, which is utilized as the key and value matrix for the cross-attention mechanism of the denoising UNet. As depicted in Fig. 6, with the incorporation of extra contextual information, the 'matmul' variant can generate more consistent and high-quality multi-view images compared to the baseline method, *i.e.,* Zero123++. Nevertheless, this variant may lead to bias in the global semantic understanding of 3D objects, such as the shapes of the bullet sample and the book sample in Fig. 6. In contrast, our method is able to generate multi-view images that are faithful to the shape and texture of objects in the input views. These experiments indicate that the LSTM modules in LSTM-GE can more effectively capture the high-level semantic information of the 3D objects.

## 5 CONCLUSIONS

In this paper, we present AR-1-to-3, a novel paradigm that starts from the input image and generates target views from near to far in an autoregressive manner. At each step of the autoregressive process, the previously generated views are employed as extra contextual information to facilitate the generation of the current target views. The experimental results demonstrate that our method generates new perspective images and 3D objects that are more consistent with the input images compared with the existing approaches generating multi-view images discretely or simultaneously. To accomplish such autoregression, we propose two image conditional strategies, *i.e.,* Stacked-LE and LSTM-GE, that encode sequence view information to fine-tune the denoising UNet model. Particularly, Stacked-LE encodes the latent features of conditional images as a stacked embedding to modify the key and value matrices of the self-attention mechanism in the UNet. Meanwhile, LSTM-GE adopts two LSTM modules to encode the CLIP features of conditional images into two feature vectors, which are concatenated together to provide high-level semantic information via cross-attention for the generation of the current target views. We hope our AR-1-to-3 could become one of the base multi-view generative models for image-to-3D, and the autoregressive idea will inspire future works.

## REPRODUCIBILITY STATEMENT

In this section, we describe the reproducibility of the results in this paper. First, we use the rendered images from the Objeaverse (Deitke et al., 2023) to train and evaluate our AR-1-to-3. We use the Blender software to render 7 images for each object based on the settings of Zero123++, including an input image and 6 target images. We also set the rendered images to a white background for the convenience of 3D object reconstruction. The overall training of our AR-1-to-3 is to learn the autoregressive paradigm, in which the views of the first k-1 steps are utilized to predict the $k$-th step views. We write a data loader to load the rendered images according to this paradigm. Second, our AR-1-to-3 is implemented using the PyTorch framework. We use the standard pipeline of the diffuser toolkit to build the Stable Diffusion network. Then, the prepared sequence data are employed to fine-tune the Stable Diffusion model equipped with our proposed Stacked-LE and LSTM-GE. The entire training process lasts for about 120k steps with a total batch size of 32 on 8 A100 GPUs. We will open-source these rendered images, source code, and our pre-trained model weights in our project to contribute 3D AI community.

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

# APPENDIX

In Sec. A, we provide more visualization examples of multi-view images synthesized by our AR-1-to-3. In Sec. B, we provide more visualization comparisons between our AR-1-to-3 and the baseline model, *i.e.,* Zero123++, on more challenge samples, including multi-object scenes and word-object scenes. These experimental results further demonstrate the superiority of our AR-1-to-3 as a base multi-view generation model for the 3D AI community.

## A VISUAL DISPLAYS

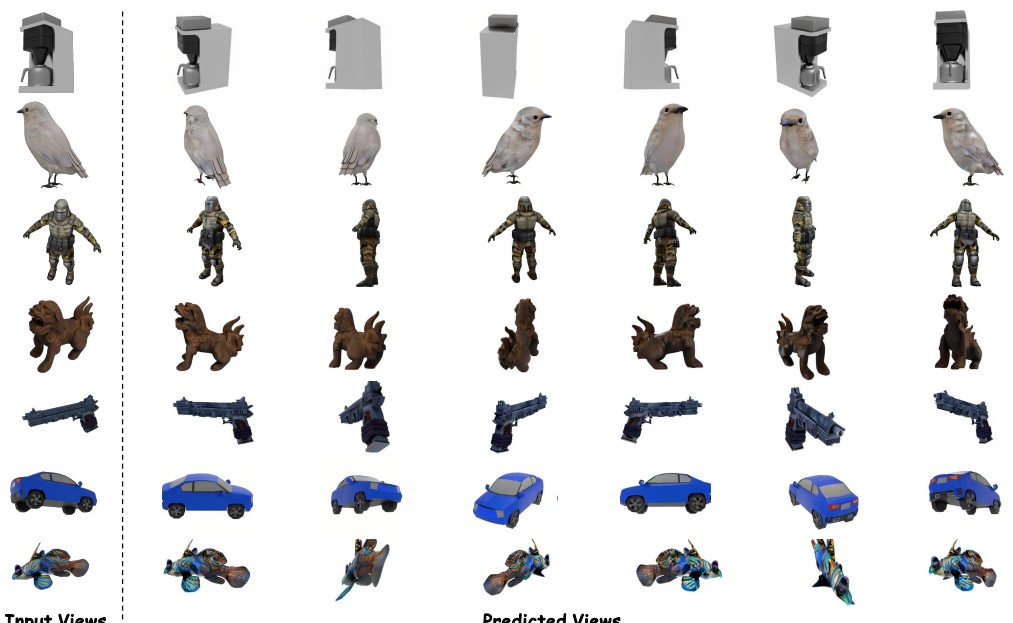

Figure 7: Multi-View Examples Generated by our AR-1-to-3.

## B VISUAL COMPARISONS

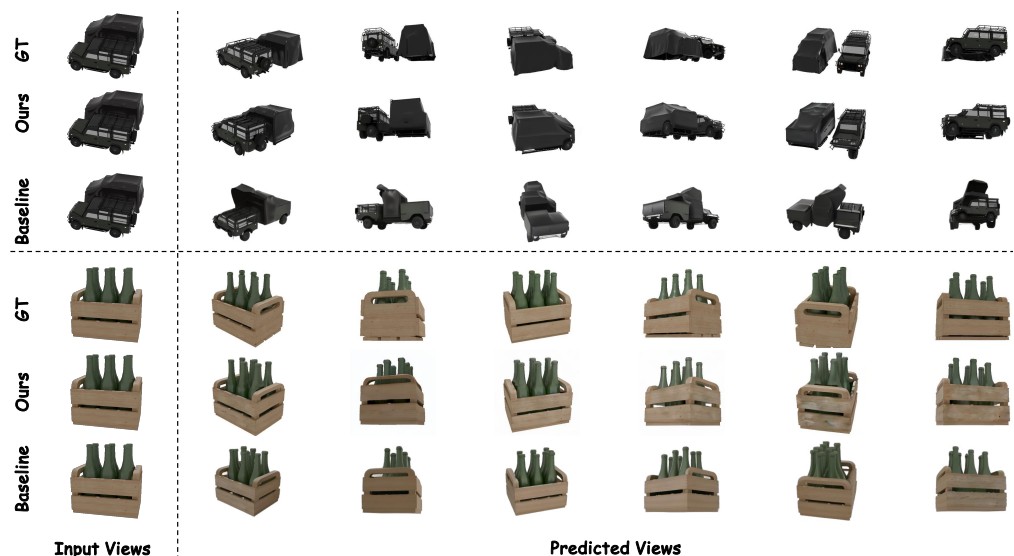

Figure 8: multi-object scenes.

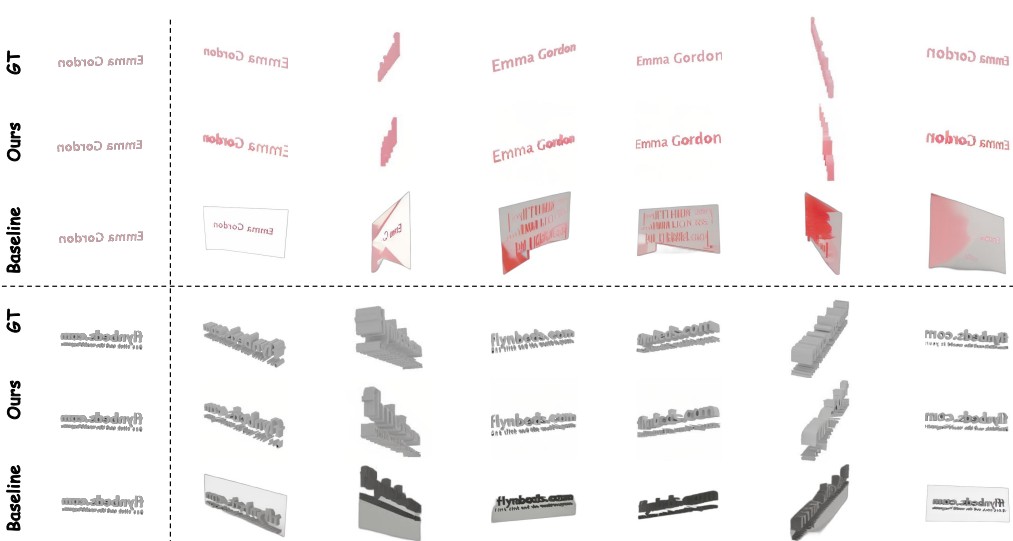

Figure 9: word-object scenes.

