# OpenReview forum: "AR-1-to-3: Single Image to Consistent 3D Object Generation via Next-View Prediction"
_ICLR.cc/2025/Conference — ICLR 2025 Conference Withdrawn Submission_

### Official Review · Reviewer_L1Cd · 2024-10-19

**Soundness:** 2
**Presentation:** 3
**Contribution:** 2
**Rating:** 5
**Confidence:** 4

**Summary:**

This paper argues that previous single-view 3D generation approaches either generate multiple discrete views of a 3D object or generate multiple views simultaneously. Both approaches will lead to inconsistency across different views and camera angles. Therefore, this paper proposes to solve this problem auto-regressively. This work will first generate views closer to the input view, which will be utilized as contextual information to prompt the generation of other views. This paper proposes two image conditioning strategies Stacked-LE and LSTM-GE to encode the image sequence. Experimental results show superior consistency of this method.

**Strengths:**

Single-view 3D generation for a single object asset is an important task and one line of research work proposes to first generate novel view images and then reconstruct the asset. This work follows this line and emphasizes tackling the inconsistency of the generated views.

1. Improving the consistency of generated multi-views is an important improvement, and the idea of solving it auto-regressively sounds novel.
2. The author proposed two techniques including Stacked-LE and LSTM-GE to encode view sequence as the local and global condition of the diffusion network, which sounds reasonable.
3. The visualized results seem to achieve superior consistency than previous methods.
4. The paper is overall well-written and should be easy to follow.

**Weaknesses:**

1. Missing related works. From the reviewer's perspective, this paper should also include other multi-view diffusion models in Sec 2.1, like MVDream[1] and ImageDream[2]. Also, since the paper targets discussing the problem of image-to-3D, another line of research work that adopts the feed-forward paradigm should also be included like LRM[3].
2. Potential baselines. As mentioned before, there are some other multi-view diffusion models. The reviewer suggests the author consider adding the comparison with ImageDream apart from Zero123++.
3. Potential zero-shot/generalization tests. Previous papers like Zero123 will evaluate zero-shot capability by testing their model on in-the-wild data like GSO or internet images. The reviewer suggests the author consider adding visualized results on other datasets or in-the-wild images. This shouldn't be hard given there are a lot of examples provided by previous works.
[1] MVDream: Multi-view Diffusion for 3D Generation, ICLR 2024

[2] ImageDream:  Image-Prompt Multi-view Diffusion for 3D Generation, arxiv 2023

[3] LRM: Large Reconstruction Model for Single Image to 3D, ICLR 2024

**Questions:**

Apart from the aforementioned weakness part, I also have the following questions.
1. Accumulation error. From the reviewer's perspective, generating views auto-regressively will cause an accumulation error. If there exists inconsistency in the initially generated two views, will the encoded feature be affected? Or should there be any designs to alleviate accumulation error if we would like to extend this auto-regressive paradigm to more complex NVS tasks in the future?
2. Potential ablation study. This paper conducted several ablative studies including sequence order, sampling ratio of latent features, and so on. While the two most important designs are the Stacked-LE and LSTM-GE, this paper does not include ablating them out during training.  So my question is what will happen if we ablate one of them during training? To what extent will they impair the performance? From the current ablative study, I am not clear on the importance of the proposed designs.
3. Potential metrics. As the method is also capable of extracting meshes, I suppose it is reasonable to also report mesh reconstruction metrics like Chamfer Distance.
4. More visualized results. I understand this kind of generation task may be hard to assess, but it will be great to see more visualized results or comparisons. Some of the examples do not seem convincing to me like the second example in Fig.8.
5. Time consumption. Both the training time and inference time can be reported in the revision.

Overall, I would like to see more visualized results, in-the-wild tests, and more ablation studies and comparisons. For now, I still have some concerns about the generalization capability and what matters the most to this method.  I would be glad to raise my rating if my concerns are addressed.

---

### Official Review · Reviewer_rBqx · 2024-10-26

**Soundness:** 3
**Presentation:** 3
**Contribution:** 3
**Rating:** 5
**Confidence:** 5

**Summary:**

The paper deals with the inconsistency problem of zero123-style NVS models. It proposes to use a procedural generation method - first generating two views that close to the input view, and then using them as the additional condition for generating further views. For this goal, the authors propose feature fusion methods for both local and global features. The proposed method is evaluated on a subset of objaverse and demonstrates better performance and consistency.

**Strengths:**

- The method involves next view prediction, which leverages additional information to reduce the uncertainty for improving the consistency of synthesized novel views.
- The method demonstrates a better performance compared with prior works (while I have some questions on it, please see weaknesses).
- The paper is well-written and is easy to follow.

**Weaknesses:**

- Baseline performance. The baseline performance in Table 1 are too low. Could the authors provide any explanation for this? Are these models trained on different renderings of objaverse so the performance is biased to the testing data which is rendered by the authors themselves, which is also used for training the model? If so, I require the authors to pick a "third-party" rendering of objaverse and evaluate the performance in a correct manner.

- Some missing related works.
1. Some recent works on improving the consistency [1,2,3,4] are missing. It would be better if the author could discuss and compare with them.
2. Using video generation model for sequential modeling. This paper is related to sequential modeling for improving the consistency. As shown in Table 2, when using shuffle viewpoints, the performance degraded significantly. However, these works using video generation model, which is another way for sequential modeling, are missing. I would suggest the authors to discuss the relation with them [5,6], e.g. whether it is auto-regression or one-shot diffusion-based generation, and potentially compare the performance.

[1] Hu, Hanzhe et al. “MVD-Fusion: Single-view 3D via Depth-consistent Multi-view Generation.” CVPR 2024.
[2] Tang, Zhenyu et al. “Cycle3D: High-quality and Consistent Image-to-3D Generation via Generation-Reconstruction Cycle.” ArXiv 2024.
[3] Weng, Haohan et al. “Consistent123: Improve Consistency for One Image to 3D Object Synthesis.” ArXiv 2023.
[4] Zheng, Chuanxia, and Andrea Vedaldi. "Free3d: Consistent novel view synthesis without 3d representation." CVPR 2024.
[5] Voleti, Vikram S. et al. “SV3D: Novel Multi-view Synthesis and 3D Generation from a Single Image using Latent Video Diffusion.” ECCV 2024.
[6] Zuo, Qi et al. “VideoMV: Consistent Multi-View Generation Based on Large Video Generative Model.” Arxiv 2024.

- Ablation experiment. I would suggest the author to ablate the use of Stacked-LE and LSTM-GE, demonstrating adding additional condition in the two blocks works.

**Questions:**

Please see the comments above.

---

### Official Review · Reviewer_viKj · 2024-10-30

**Soundness:** 3
**Presentation:** 2
**Contribution:** 2
**Rating:** 5
**Confidence:** 4

**Summary:**

The paper introduce AR-1-to-3, an approach for generating multi-view images from a single input image with enhanced detail consistency. This is achieved through an auto-regressive scheme where new views are generated based on previously generated ones. Unlike other previous methods, the approach first generates views closer to the input, using these as context for generating farther views. The paper propose two image conditioning strategies, Stacked-LE and LSTM-GE, to encode the sequence of views. Qualitative and quantitative results are shown to present the effectiveness of the model.

**Strengths:**

1. Sufficient ablation analysis. The paper presents sufficient ablation studies to explore the effectiveness of generation order, sampling ratio and encoding strategy.
2. Good performance shown in qualitative and quantitative results.

**Weaknesses:**

1. Unfair comparison in Table 1. The metrics in Table 1 are used to evaluate the quality of generated images. Although SyncDreamer cannot generate the specific poses as AR-1-to-3's, it is unfair to compare with the renderings of reconstructed meshes. Because the step of reconstruction and then rendering will reduce the quality of the images.
2. Fixed generation poses and numbers restricts the ability of the model to fully utilize its capabilities. Leveraging autoregressive scheme, it is natural to generate arbitrary poses or numbers of novel views. Otherwise, there is no substantial benefit in adopting this approach, which consumes more resources and time. Could you please discuss the potential for extending your method to generate arbitrary poses or numbers of views, and to analyze the trade-offs between flexibility and computational cost in your approach.
3. Insufficient evaluation dataset. Evaluating only on Objaverse is limited, and more testing datasets should be evaluated, such as Scanned Objects by Google Research.

**Questions:**

1. Will this form of autoregression encounter the problem of error accumulation? Have you dealt with such a situation.
2. Why choose to generate two images at a time instead of one? Is it a trade-off between time and effect? Is there any analysis in this regard?
3. AR-1-to-3 far surpasses Zero123++ in Table 1. What key factors have brought about such significant improvements?

---

### Official Review · Reviewer_vmHg · 2024-11-03

**Soundness:** 3
**Presentation:** 3
**Contribution:** 3
**Rating:** 5
**Confidence:** 5

**Summary:**

This paper addresses the problem of object-centric novel view synthesis from a single or sparse  input image. This paper proposes a novel AR diffusion approach with hybrid-architecture (block-wise casual attention and LSTM). The authors conduct experiments on Objverse and report quantitative results on it, outperforming other recent works. They also investigate the impact of synthesis order, demonstrating improved results when compared to random and reverse order generation. This AR diffusion approach might be applied to lots of other problems.

**Strengths:**

The AR diffusion algorithm is innovative and potentially applicable to a range of tasks, including long video generation.

The wirting is clear, and I can understand the technical part quite easily.

Also the author showed state of the art results on Objverse (Synthetic Dataest).

**Weaknesses:**

There are a key design choices and questions in AR-diffusion which are not closely disscused.

1. **Noise strategy**: The authors mention adding the same noise timestep to both reference and target frames. However, there is no explanation or ablation study to justify this choice. Why was this approach chosen over others? For example, you can keep reference frames as clean images, and only add noise to target denosing views. Or you can add indenpendent noise levels, or using increasing noise levels for target views.

2. **Efficient Parallel Training Across Target and References frames**.   For example, when using an input image and reference views 0 and 1 to denoise target views 2 and 3, is the loss calculated only for target views 2 and 3? With block-wise causal attention and the same-noise-level strategy, it seems possible to calculate diffusion loss for reference views (e.g., views 0 and 1) as well. How would this impact the results? Idealy, this seems more efficient, right? Like in LLM nowdays, decoder-only arch can compute loss at every tokens, but Bert-based approach can only compute loss at MASK tokens, which is not that efficient (regarding the number of loss tokens divided by the number of FLOPS).



Another comments, which might seems objective. The authors seems to suggest that AR diffusion might produces more 3D consistent results compared to full sequence diffusion ones. Though, when doing full sequence diffusion, the target frames cannot extract features from clean reference frames (because reference frames are at the same noise level as current frames), this claim remains unconvincing for me, without direct comparisons using the same model architecture, data, and computational resources.



I think it's better to add some citations for other AR-diffusion paper, though not applied to the same task, like DiffusionForcing, Rolling Diffusion Models, and Pyramid Flow (Came out on Oct, you don't need to cite it now, but recommend to cite later.).

**Questions:**

1. The evaluation is conducted only on the synthetic dataset Objaverse. Given that ABO and GSO are commonly used benchmarks in this field, could you provide additional results on these datasets?
2. Several ablations are missing. For instance, what is the impact of the LSTM on model performance and inference speed? Why is the noise added to the reference frames the same as for the target frames?
3. Do you observe any error accumulation effects? AR algorithms trained with teacher forcing often suffer from error accumulation. Out of curiosity, have you observed increasing artifacts in later synthesized views? or maybe 6 views is not long enough to observe strong error accumulation artifacts.
4. Regarding Figure 5, does the reported FLOPs represent the entire model or just the attention module? If it's for the attention module alone, could you also provide the total model FLOPs? This would help in understanding the extent of computational savings achieved by the alpha parameter in a single evaluation (also, consider mentioning if KV-cache can be used when generating new views).

---

### Note · Authors · 2024-11-22

I have read and agree with the venue's withdrawal policy on behalf of myself and my co-authors.